# Intelligent Compound Fault Diagnosis of Roller Bearings Based on Deep Graph Convolutional Network

**DOI:** 10.3390/s23208489

**Published:** 2023-10-16

**Authors:** Caifeng Chen, Yiping Yuan, Feiyang Zhao

**Affiliations:** School of Mechanical Engineering, Xinjiang University, Urumqi 830047, China; ccf@stu.xju.edu.cn (C.C.); chencaifengxj@163.com (F.Z.)

**Keywords:** intelligent compound fault diagnosis, deep graph convolutional neural network, roller bearing

## Abstract

The high correlation between rolling bearing composite faults and single fault samples is prone to misclassification. Therefore, this paper proposes a rolling bearing composite fault diagnosis method based on a deep graph convolutional network. First, the acquired raw vibration signals are pre-processed and divided into sub-samples. Secondly, a number of sub-samples in different health states are constructed as graph-structured data, divided into a training set and a test set. Finally, the training set is used as input to a deep graph convolutional neural network (DGCN) model, which is trained to determine the optimal structure and parameters of the network. A test set verifies the feasibility and effectiveness of the network. The experimental result shows that the DGCN can effectively identify compound faults in rolling bearings, which provides a new approach for the identification of compound faults in bearings.

## 1. Introduction

Bearings are an important part of rotary equipment and are also the most easily damaged component. According to statistics, in rotating machinery using rolling bearings, about 30% of mechanical failures are related to bearings [1], and the operating condition directly affects the overall performance of the equipment [2]. In bearing fault diagnosis, the extraction of fault characteristic information from the original vibration signal still plays a dominant role [3]. However, in actual engineering, a variety of bearing failures co-exist to form compound failures. The vibration signals of a compound fault condition are not simply a superposition of single fault signals but are coupled with the vibration signals of other components through complex transmission paths. The faults of different components interact and influence each other, causing the composite fault signal to be characterized by non-smoothness and non-linearity [4] and bringing great difficulties in bearing fault diagnosis. Therefore, how to extract all kinds of fault characteristics from the composite fault signal is the focus and difficulty of current research.

The most important step in the traditional rolling bearing fault diagnosis is feature extraction. The purpose is to extract useful fault information in the signal so as to improve the accuracy of fault diagnosis [5]. Common feature extraction methods include Wavelet Transform (WT), Principal Component Analysis (PCA), Short Time Fourier Transform (STFT), etc. [6]. Currently, most fault diagnosis methods require manual feature selection when dealing with non-smooth, non-linear signals, and different feature selections determine the effectiveness of fault diagnosis. On the other hand, the performance of existing feature extraction methods gradually decreases as the amount of data increase.

Deep learning theory is introduced into the field of fault diagnosis to overcome the above shortcomings. For example, Jiang et al. [7] proposed a multi-scale convolutional neural network (MSCNN) to automatically learn fault characteristics from the original vibration signal and classify gearbox fault types. Xu et al. [8] used deep convolutional neural networks (DCNN) to directly process the original vibration signals, thus realizing the fault diagnosis of rolling bearings. Compared with the traditional shallow model, deep learning builds an “end-to-end” model. Health information can be obtained directly from the collected signals, while meeting the high requirements for fault diagnosis accuracy in the era of big data [9]. The above deep learning method can only learn features from the vertices of the input data, while ignoring the information contained in the edges formed between vertices.

Inspired by this, graph neural networks (GNNs) were introduced to extend CNNs to graph data [10,11] to solve the above problem. The GNN was first proposed by Scarselli et al. in 2009 [12] and is based on graph theory to construct a neural network for data in the graph domain. In the graph domain, using the attributes of nodes and edges can provide additional information to improve the extracted features. In addition, the noise immunity has been improved. As a result, more information can be provided in a graphics field than in a general data field. Bruna et al. [13] introduced convolutional operations to GNNs which are based on spectral graph theory and constructed the first graph convolutional network (GCN) model. Compared with traditional CNN methods, GCN has advantages in dealing with non-stationary, non-linear signals and discriminative feature extraction of discrete spatial domain signals [14]. Up to now, the GCN method has been successfully applied to several research areas such as intelligent acoustic fault diagnosis of rolling bearings [15] and wind turbine gearbox fault diagnosis [16,17,18]. The main research focus of this paper is how to construct the original vibration signal into graph-structured data. Secondly, the GCN model’s feature mining capability is utilized to eliminate the screening feature link in fault diagnosis and improve the efficiency of composite fault diagnosis.

The structure of this paper is as follows. In Section 2, the theory of the work in question is presented. Section 3 describes the process for composite bearing fault diagnosis based on deep graph convolutional networks. The feasibility of the proposed method is verified in Section 4. The conclusion is presented in the Section 5.

## 2. Related Work

The core idea of GCN is to extend convolutional operations from Euclidean datasets to non-Euclidean datasets by aggregating information about the nodes in the graph and their neighbors via means of supervised or semi-supervised learning. Thereby, extracting high-dimensional features in the graph structure allows for numerous tasks such as node prediction, classification and edge prediction. The GCNs can be divided into spectral and space GCNs. The use of a spectrum-based GCN in this article mainly involves three steps: First, a graph Fourier transform is applied to the input data. Secondly, the results of the transformation are convolved in the spectral domain. Finally, the convolution results are subjected to an inverse graph Fourier transform.

### 2.1. Representation of Graphs

This study defines the original data into sub-samples based on graph theory as a graph with nodes and edges, which for any undirected graph can be represented as:(1)$$\left\{\begin{array}{l} G=\left(V,E,A\right)\;\;\;\;\;\;\;\;\;\;\;\;\;\\ V=\left(v_{0},v_{1},\cdots ,v_{n}\right)\;\;\\ E=\left(v_{i},v_{j}\right)\;\;\;\;\;\;\;\;\;\;\;\;\;\;\; \end{array}\right.$$
where *V* refers to the set of *N* nodes, *E* refers to the set of edges, *A*∈*R^N^*
^× *N*^ is the adjacency matrix defining the interconnections between nodes and in the undirected graph, *A_i_*_,*j*_ = *A_j_*_,*i*_.

Other than the adjacency matrix *A*, the graph can be defined as a Laplacian matrix with *L* = *D* − *A*, where *D* and *A* denote the degree matrix and the adjacency matrix, respectively. The adjacency matrix and degree matrix are calculated as shown in Figure 1. The degree matrix indicates the number of connected nodes. For example, with node one having one edge and node five having four edges. The adjacency matrix represents the relationship between nodes. For example, node one and node five are connected and represented by one, while node one and node two are not connected and represented by zero.

### 2.2. Spectrogram Convolution

In spectral graph convolution, the symmetric normalized graph Laplace operator is usually used and is defined as:(2)L=IN−D−12AD−12
where $D=diag\left(\sum_{j}A_{ij}\right)$ refers to the degree matrix and $I_{N}$ is the unit matrix.

The graph Laplacian matrix is a real symmetric matrix whose eigenvalues can be decomposed as:(3)$$L=U\wedge U^{-1}=U\left[\begin{array}{c} \lambda_{\begin{array}{c} 1\;\;\;\;\;\;\;\;\;\\ \ddots \end{array}}\\ \;\;\;\;\;\;\;\;\;\lambda_{n} \end{array}\right]U^{-1}$$
where Λ is a diagonal matrix of eigenvalues and *U* is a matrix of eigenvectors.

The spectral convolution of the set of nodes *V* with the node features can be expressed as:(4)$$h={\left(x{*}_{G}f\right)}_{\theta }=U\left(\left(U^{T}x\right)⊙\left(U^{T}f\right)\right)$$
where *h* refers to the feature map after graph convolution, **G* represents the graph convolution, *x* is the node feature, *f* is the feature function of Λ, i.e., *f*(Λ), *θ* is the learnable parameter, ⊙ is the Hadamard product of elemental forms and *U^T^ x* refers to the graph Fourier transform of the node feature *x*.

Using *f_θ_* = *U^T^f* as a learnable graph convolution filter, the above equation simplifies to:(5)$$h=Uf_{\theta }U^{T}x.$$

However, the filter *f_θ_* is computationally complex and not spatially localized. This article uses the Chebyshev polynomials to approximate the filter and the derived graph convolution and making λmax ≈ 2 further simplifies the Chebyshev polynomials, which can be expressed as:(6)$$f_{\theta }=\sum_{k=0}^{K}\theta_{k}T_{k}\left(\tilde{\wedge }\right)$$
(7)$$h=\theta_{0}x+\theta_{1}\left(L-I_{n}\right)x=\theta_{0}x-\theta_{1}D^{-1/2}AD^{-1/2}$$
where *K* stands for Chebyshev’s polynomial order, $\wedge =dig\left(\left[\lambda_{0},\lambda_{1},\cdots ,\lambda_{n-1}\right]\right)$ and $\lambda_{n-1}$, respectively, represent the eigenvalues array and eigenvalues for *L*, $\tilde{\wedge }=2\wedge /\lambda_{max}-I_{n}$ refers to the rescaled eigenvalue matrix and *T_k_* refers to the Chebyshev polynomial.

Let *θ* = *θ*_0_ = −*θ*_1,_ then the above equation becomes:(8)$$h=\theta \left(I_{n}+D^{-1/2}AD^{-1/2}\right)x.$$

To alleviate the gradient explosion/disappearance problem, $I_{n}+D^{-1/2}AD^{-1/2}$ is further reduced to $D^{-1/2}AD^{-1/2}$ and the final expression is:(9)$$H=\sigma \left(D^{-1/2}AD^{-1/2}X\Theta \right)$$
where *H* is the convolutional signal matrix, *σ* refers to the non-linear activation function and *Θ* refers to the learnable parameters.

### 2.3. Graph Convolutional Networks

Following is a non-linear activation function, where the GCN with a single message passing is expressed as:(10)$$H^{\left(l+1\right)}=\sigma \left(D^{-1/2}AD^{-1/2}H^{\left(l\right)}W_{1}^{\left(l\right)}+H^{\left(l\right)}W_{0}^{\left(l\right)}+b^{\left(l\right)}\right)$$
where $H^{\left(1\right)}\in R^{N\times d_{1}}$ is the hidden matrix of nodes of dimension *d*_l_ at level l, $H^{\left(0\right)}=X$ refers to the matrix of input node characteristics, *σ*(·) refers to the ReLU activation function, $W_{0}^{\left(l\right)}\in R^{d_{l}\times d_{l+1}}$ and $W_{1}^{\left(l\right)}\in R^{d_{l}\times {\;d}_{l+1}}$ denote the learnable parameter matrix and $b^{\left(l\right)}$ is the bias vector. A GCN with a message passing operation can be considered as a first-order approximation to a spectral map convolution. In order to further reduce the model parameters, the single-parameter GCN model at layer l can be expressed as:(11)$$H^{\left(l+1\right)}=\sigma \left({\tilde{D}}^{-1/2}\tilde{A}{\tilde{D}}^{-1/2}H^{\left(l\right)}W^{\left(l\right)}+b^{\left(l\right)}\right)$$
where $\tilde{A}=A+I_{n}$, ${\tilde{D}}^{-1/2}\tilde{A}{\tilde{D}}^{-1/2}$ indicates that a self-connected normalised adjacency matrix has been added and ${\tilde{D}}_{ii}=\sum_{j}{\tilde{A}}_{ij}$ and $W^{\left(l\right)}\in R^{d_{l}\times d_{l+1}}\;$ is the learnable parameter matrix. GCN reduces over-fitting by using a first-order approximate filtering operation that involves fewer free parameters per filtering operation.

## 3. Compound Fault Diagnosis of Bearings Based on Deep Graph Convolution

This article proposes a composite fault diagnosis method for bearings based on deep graph convolutional networks, with the process shown in Figure 2. Firstly, vibration signals from bearings are collected and divided into sub-samples. Secondly, the sub-samples will form the sample map. Finally, the DGCN model is used to extract the features of the graph and achieve rapid classification of composite faults.

### 3.1. Building Graphs Based on Time Series

The original time series *X* with signal length *L* is normalized and denoted as:(12)$$X^{nol}=normalization\left(X\right)$$
where *X^nol^* is the normalized time series and normalization(∙) refers to a different normalization method. Max-min normalization is used in this paper.

Next, the signal length *L* original time sequence is divided into sub-samples of length *d*. There is no overlap between each sample and a corresponding label is assigned to each sub-sample. The obtained sub-sample set can be expressed as:(13)$$\prod =\left[\left(x_{1}^{nol},y_{1}\right),\left(x_{2}^{nol},y_{2}\right),\cdots ,\left(x_{n}^{nol},y_{n}\right)\right]$$
(14)$$n=\left[\frac{L}{d}\right]$$
where ∏ refers to the constructed super-sample set, $x_{n}^{nol}$ refers to the sub-sample and $y_{n}$ refers to the label. *n* refers to the number of sub-samples.

Drawing on reference [19], the use of frequency domain inputs can improve the performance of the model. Therefore, the author performs a Fast Fourier Transform (FFT) on each sub-sample and uses the transformed data as a new sample. The process can be expressed as:(15)$${\tilde{x}}_{i}=FFT\left(x_{i}^{nol}\right),i=1,2,\cdots ,n$$
where FFT(∙) refers to converting the sub-sample to the frequency domain and taking the first half of the result.

The resulting labeled dataset is represented as:(16)$$\tilde{\prod }=\left[\left({\tilde{x}}_{1},y_{1}\right),\left({\tilde{x}}_{2},y_{2}\right),\cdots ,\left({\tilde{x}}_{n},y_{n}\right)\right].$$

### 3.2. Structural Data for Construction Diagrams

The author uses the radius graph method to construct graph structure data. The specific methods are as follows: Firstly, the *n* sub-samples obtained above are used as the *n* nodes of the graph. Secondly, cosine similarity is used to estimate the distance between each sub-sample and set a threshold ε. If the cosine similarity is larger than the threshold, there will be an edge between the two knots. Therefore, the neighbors of node *x_i_* can be obtained by:(17)$$Ne\left({\tilde{x}}_{i}\right)=\left\{\begin{array}{l} \varepsilon -\mathrm{radius}\left({\tilde{x}}_{i},y_{i}\right),\;\mathrm{if}\;\;s\left({\tilde{x}}_{i},{\tilde{x}}_{j}\right)>\varepsilon \\ 0,\;\;\;\;\mathrm{otherwise}\;\;\;\;\;\;\;\;\;\;\;\;\;\;\;\;\;\;\;\;\;\;\;\;\;\;\;\;\;\;\;\;\;\;\; \end{array}\right.$$
where $Ne\left({\tilde{x}}_{i}\right)$ is the neighbor node of ${\tilde{x}}_{i}$, *ε* indicates the selected radius, here ε is 0 and $\left({\tilde{x}}_{i},{\tilde{x}}_{j}\right)$ calculates the cosine similarity of nodes ${\tilde{x}}_{i}$ and ${\tilde{x}}_{j}$.

The weight between each two nodes is calculated by a threshold Gaussian kernel weight function, which is expressed as:(18)$$w_{ij}=\left\{\begin{array}{l} \exp\left(-\frac{s^{2}\left({\tilde{x}}_{i,}{\tilde{x}}_{j}\right)}{2\beta^{2}}\right),\;\mathrm{if}\;\;s\left({\tilde{x}}_{i},{\tilde{x}}_{j}\right)>\varepsilon \\ 0,\;\;\;\;\mathrm{otherwise}\;\;\;\;\;\;\;\;\;\;\;\;\;\;\;\;\;\;\;\;\;\;\;\;\;\;\;\;\;\;\;\;\;\;\; \end{array}\right.$$
where *β* denotes the bandwidth variance of the Gaussian function.

### 3.3. Fault Diagnosis Structure

In this article, the proposed DGCN model includes seven layers, one input layer, two graph convolution layers, two BatchNorm layers, one fully connected layer and one output layer. Among them, graph convolution layers are used to extract features of nodes and edges. The BatchNorm layer enables faster and more stable training of edges and nodes; the fully connected layer is used for node classification.

## 4. Case Study

### 4.1. Description of Experimental Data

In this section, the XJTU-SY [20] bearing dataset is used to verify the validity of the proposed DGCN method. As shown in Figure 3, the experimental platform mainly consists of a drive motor, support shaft, speed controller, support bearings, test bearings and hydraulic loading system. The test bearing is LDK UER204 rolling bearing. The sampling frequency was 25.6 kHz and the sampling interval was one minute. The experiments selected in this paper are shown in Table 1, and in this paper, six bearings were selected for two operating conditions, with a speed of 2100 r/min for condition one and 2400 r/min for condition three. Each condition contains one normal and three faults as shown in Figure 4.

The vibration signals collected from the XJTU-SY test stand are shown in Figure 5 and are pre-processed. First, the original signal was subjected to a max-min normalization process. Second, a sliding window was used to intercept non-overlapping vibration signals, with each sub-sample containing 1024 data points and 1000 samples for each fault type, for a total of 8000 samples for the eight fault types. Finally, each sub-sample was FFT transformed and labeled accordingly. The division of the training and test sets is shown in Table 2. Dataset A contains two working conditions, the fault types of condition one include normal, outer ring fault, cage fault, and composite fault of inner and outer rings. The fault types of condition three include normal, outer ring failure, inner ring, rolling element and cage composite failure, and inner ring failure. Diagnosis of faults in the two operating conditions was conducted separately, with 80% of the samples from each bearing health condition being used for training and the rest for testing in order to compare the same working conditions with different working conditions. In Dataset B, the data for the different working conditions were considered as a whole and only one normal bearing was considered, so that there were seven health states in total. 80% of the samples for each bearing health state were used for training and the remaining samples for testing. In practice, compound fault samples were more difficult to collect than single fault samples. Thus, in dataset C, the percentage of training samples for normal bearings was 50%, the percentage of training samples for single faults 30% and the percentage of training samples for compound faults was 20% or 10%.

### 4.2. Constructing the Graph

This study selected eight fault samples for two operating conditions, each one with a signal length of 1,024,000 and a sub-sample length of 1024, for a total of 1000 sub-samples. One every ten sub-samples was used to form one ε-radius of zero, and 2β2 is a graph of 1024 as shown in Figure 6. First, calculate the cosine similarity of each sample to its ten nearest neighbors and obtain the weights of the edges by Equation (18). Then, move to the next ten neighbors until all samples are traversed. Finally, move to the next ten neighbors’ nodes until all samples are traversed.

The adopted approach is effective in extracting graph features. Node information is embedded in each layer and fault features in the original data are extracted by the graph convolution layer. This study is a classification of nodes. Therefore, the input layer and the structure of the network does not change after one layer of convolution and two layers of convolution. As shown in Figure 7, each layer of the network is visualized by Gephi.

### 4.3. Detailed Framework for Fault Diagnosis

This paper employs the use of momentum stochastic gradient descent (SGD) as an optimizer for hyperparameter optimization, where the momentum of SGD is 0.9, the number of iterations is 100 and the batch size is 64. The learning rate decay strategy also adjusts the learning rate, where the weight decay value is initialized at 0.0005, the learning rate is 0.001, the number of network nodes is 640 and the number of edges is 5760. The network connection of the model is I-C1-B1-C2-B2-FC1-FC2, where C stands for the convolutional layer, B stands for the normalization layer and FC stands for the fully connected layer. The network structure for the specific model is set out in Table 3.

### 4.4. Results and Analysis

Each experiment was repeated 10 times to reduce the randomness of the results and then the average of the 10 test results was taken as the final result.

#### 4.4.1. Compound Fault Diagnosis under the Same Operating Conditions

This section focuses on the performance of the proposed DGCN in terms of classification under the same operating conditions and with balanced samples. Therefore, this section considers Dataset A_1 and Dataset A_2. Figure 8 shows the t-SNE visualization results of the fault features extracted by GCN. It can be seen from the graph that this method is able to extract useful fault features under the same operating conditions and all fault types can be clearly distinguished, achieving good fault detection accuracy. The confusion matrix of the diagnostic results is shown in Figure 9, indicating that the DGCN model can effectively detect multiple faults under the same operating conditions.

#### 4.4.2. Compound Fault Diagnosis under Different Operating Conditions

For comparison, this section used the proposed DGCN for fault classification under different operating conditions, therefore considering Dataset B. The extracted features are also visualized using t-SNE. As shown in Figure 10, it can be seen that the method can effectively extract composite fault data features. The confusion matrix of the classification results is shown in Figure 11. It can be found that the method can effectively avoid the problem of mutual interference between single faults and compound faults leading to more misclassifications. This shows that the proposed method is effective in compound fault diagnosis.

#### 4.4.3. Diagnosis of Compound Faults under Sample Imbalance

In fact, fault samples are harder to collect than normal samples, while composite fault samples are more difficult to collect than single fault samples. In this section, we used the proposed DGCN to classify the imbalanced samples and therefore consider Dataset C. The predictions of the model on the test set were visualized using t-SNE, as is shown in Figure 12, and the characteristics of the different fault types are well separated. Therefore, DGCN is able to learn better features from vibration signals. It can be seen from Figure 13 that DGCN has a very high accuracy in all the confusion matrices. These results demonstrate the superiority of this method in the classification of unbalanced samples.

### 4.5. Evaluation Indicators

It is worth noting that the following three evaluation metrics were chosen in order to provide a comprehensive assessment of the model’s performance for different datasets: overall accuracy (*ACC*), true positive rate (*TPR*) and false acceptance rate (*FAR*). Each experiment was conducted 10 times to reduce the effect of randomness, as defined below.
(19)$$ACC=\frac{TN+TP}{TN+TP+FN+FP}$$
(20)$$TPR=\frac{TP}{TP+FN}$$
(21)$$FAR=\frac{FP}{TN+FP}$$
where *TP* refers to a correctly judged unstable sample, *TN* refers to a correctly judged stable sample, *FP* refers to a misjudged unstable sample and *FN* refers to a misjudged stable sample.

The performance metrics on the test set under different datasets are shown in Table 4. It can be found that the overall accuracy of DGCN is high, and the performance on stable and unstable samples is relatively balanced. This fully demonstrates that the model is capable of achieving compound fault classification.

The author also investigated the effect of different distance calculation methods and the number of nodes on the accuracy of the change algorithm, and the results are as follows.

The effect of the number of nodes on the accuracy of the different datasets is shown in Figure 14. It can be found that in the same working condition dataset (DatasetA_1 and DatasetA_2) testing, the accuracy rate basically tends to be stabilized when the number of nodes is 5–10 but decreases significantly when the number of nodes is greater than 15. In cross working conditions and imbalanced sample dataset testing, changes in the number of nodes had little change in accuracy.

Figure 15 shows the impact of using the three distance calculation methods on the diagnostic accuracy of the different datasets. Using cosine similarity to calculate distance has the highest accuracy. In contrast, the accuracy of Euclid and Chebyshev distance calculation methods is slightly less accurate.

## 5. Conclusions

This paper proposes a composite fault diagnosis method for rolling bearings based on deep graph convolutional networks. This method solves the shortcomings of traditional fault diagnosis methods such as needing to extract features manually and relying on expert knowledge. We can be obtain the following conclusions from the experiments:(1)The DGCN model enables an end-to-end diagnostic model, eliminating the need for complex feature engineering in traditional diagnostic methods. Graphs constructed using vertices and edges can provide more information for the training of diagnostic models. Thus, the experimental results indicate that the DGCN method is highly advantageous in identifying rolling bearing faults with different operating conditions and sample imbalances.(2)To address the problem of high correlation between compound faults and single fault samples that can easily cause misclassification, this experiment proves that DGCN can effectively avoid misclassification between single faults and compound faults.

However, this study has not been combined with the bearing mechanism, so its interpretability is not strong. In addition, this study used a standard dataset and did not have a strong generalization ability for actual running data.

## Figures and Tables

**Figure 1 sensors-23-08489-f001:**
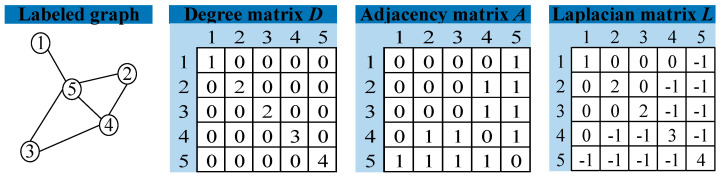
Laplace matrix calculation.

**Figure 2 sensors-23-08489-f002:**
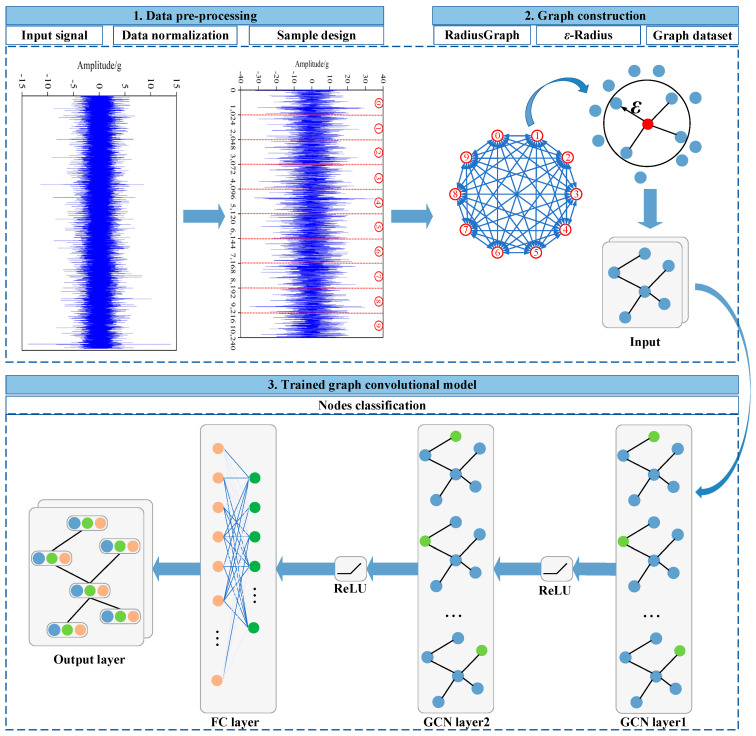
A framework for composite fault diagnosis of bearings based on depth map convolution.

**Figure 3 sensors-23-08489-f003:**
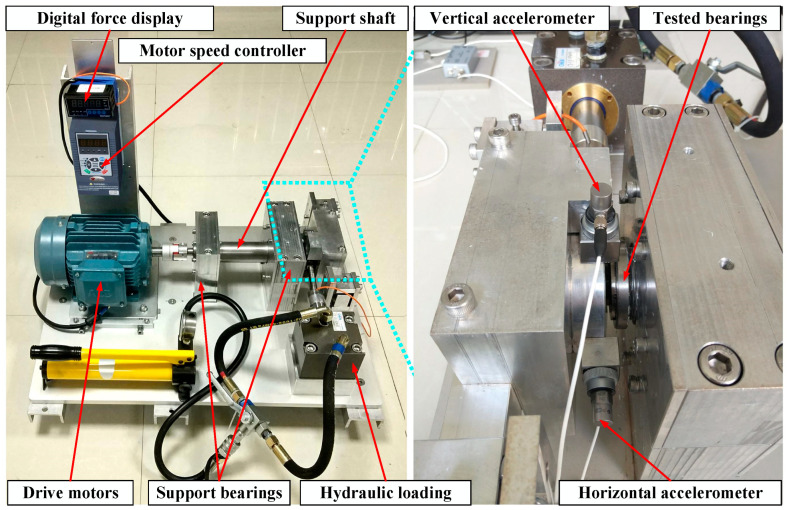
Bearing tested.

**Figure 4 sensors-23-08489-f004:**
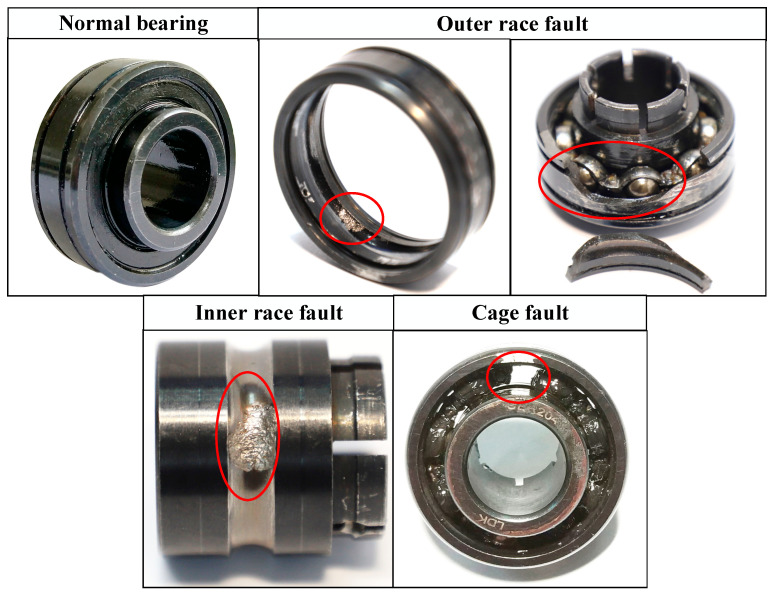
Photos of tested bearings.

**Figure 5 sensors-23-08489-f005:**
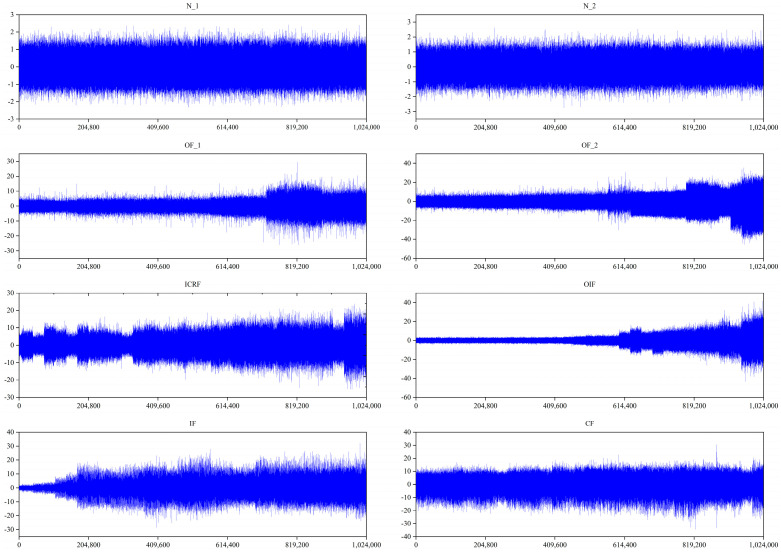
Vibration signals for each health condition of the bearing.

**Figure 6 sensors-23-08489-f006:**
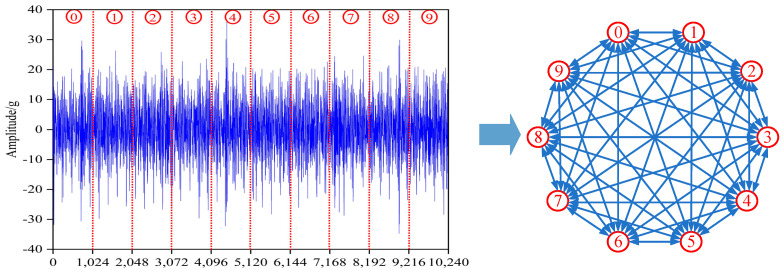
Graph data construction process. (The raw signal is divided into 10 parts, each representing a node).

**Figure 7 sensors-23-08489-f007:**
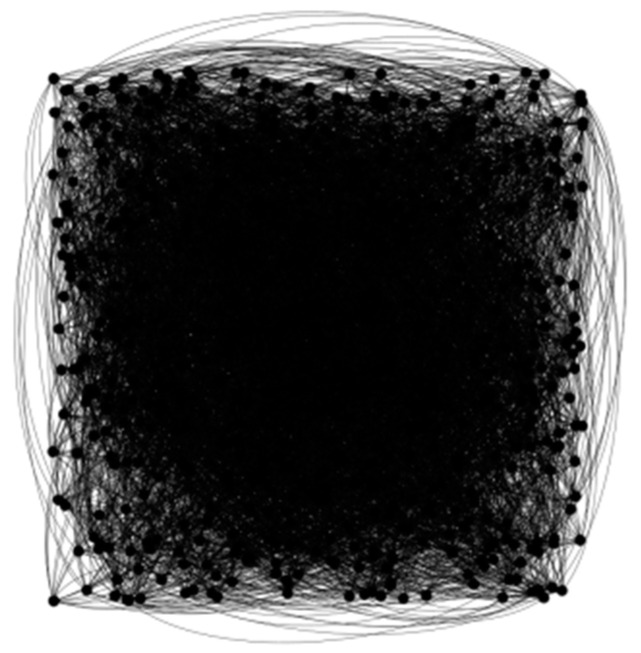
Graph Structure Visualization.

**Figure 8 sensors-23-08489-f008:**
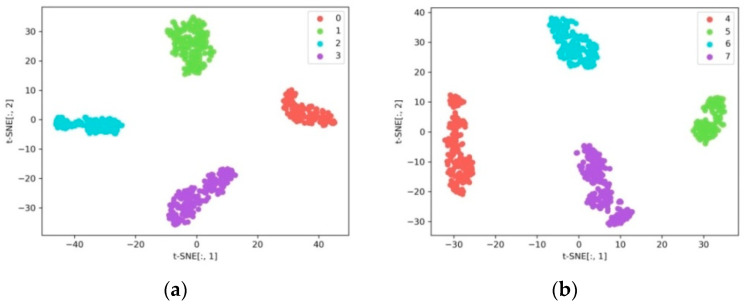
Results of feature visualization under the same working conditions. (**a**) Dataset A_1, (**b**) Dataset A_2.

**Figure 9 sensors-23-08489-f009:**
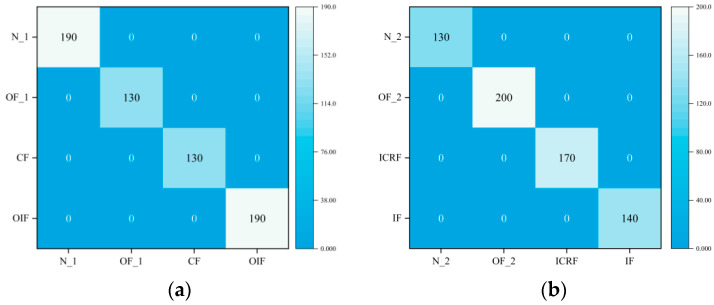
Confusion matrix for the same operating conditions. (**a**) Dataset A_1, (**b**) Dataset A_2.

**Figure 10 sensors-23-08489-f010:**
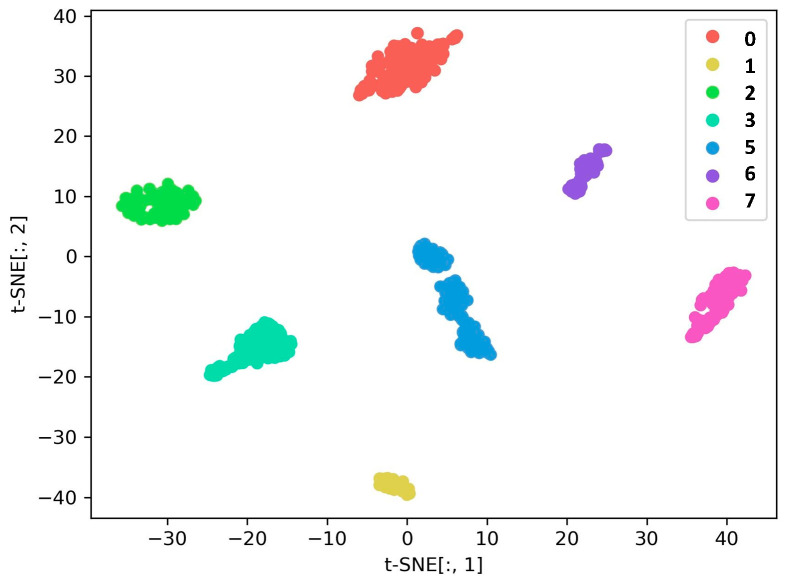
Results of feature visualization under different working conditions.

**Figure 11 sensors-23-08489-f011:**
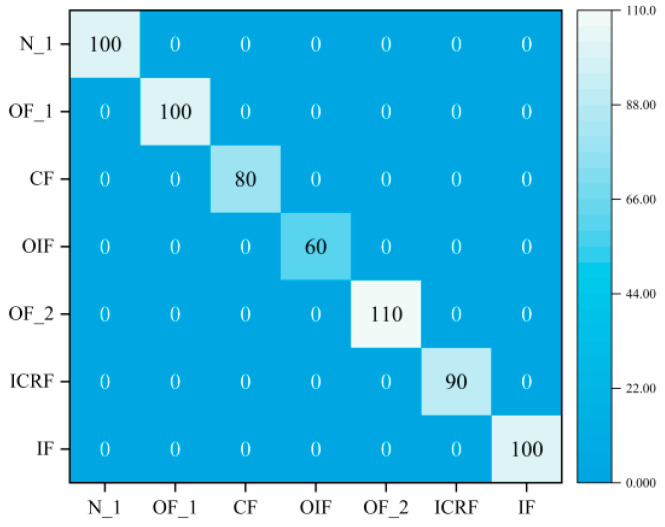
Confusion matrix for different operating conditions.

**Figure 12 sensors-23-08489-f012:**
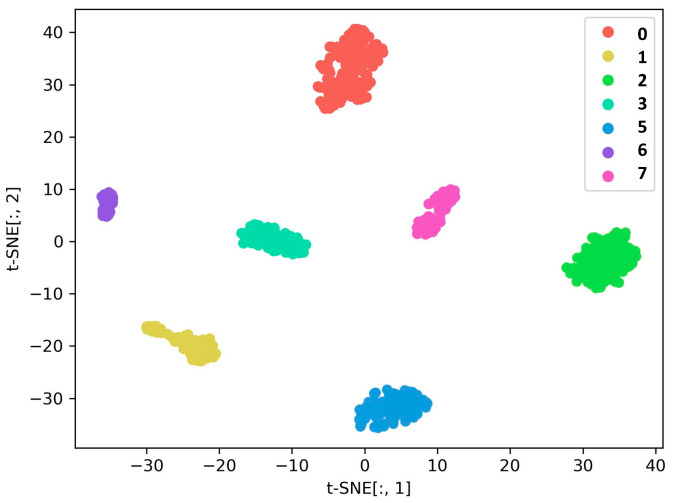
Results of feature visualization under sample imbalance.

**Figure 13 sensors-23-08489-f013:**
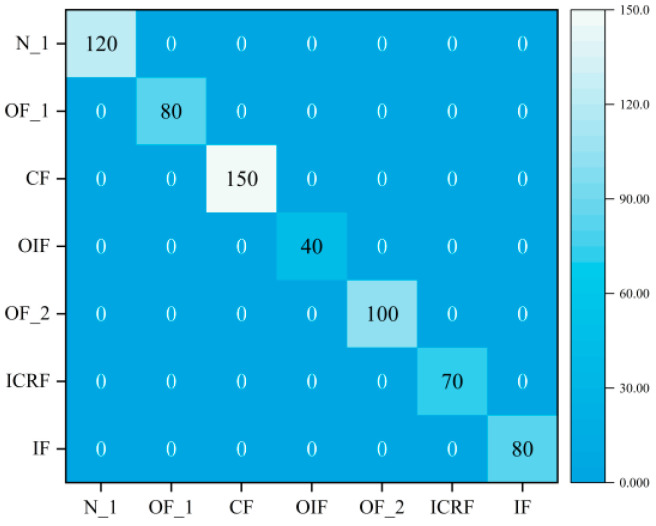
Confusion matrix under sample imbalance.

**Figure 14 sensors-23-08489-f014:**
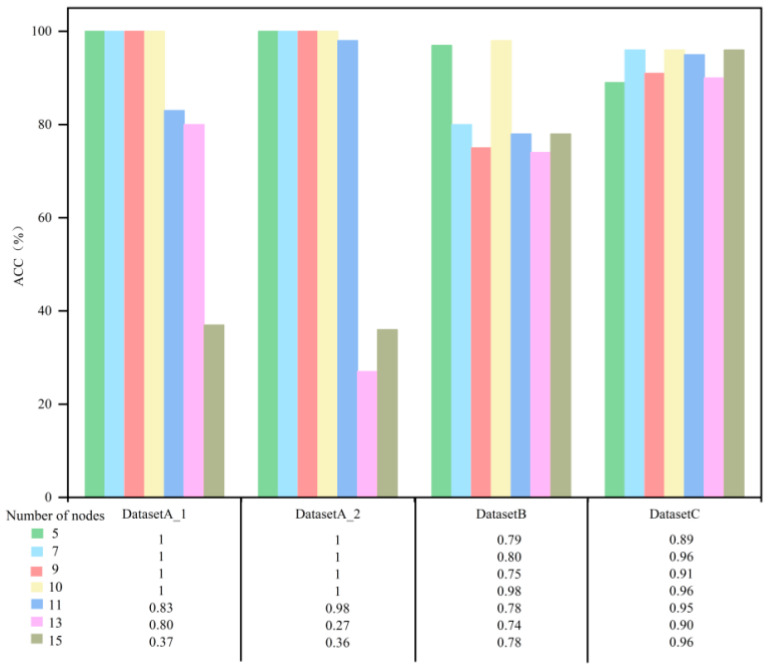
Effect of the number of nodes on accuracy.

**Figure 15 sensors-23-08489-f015:**
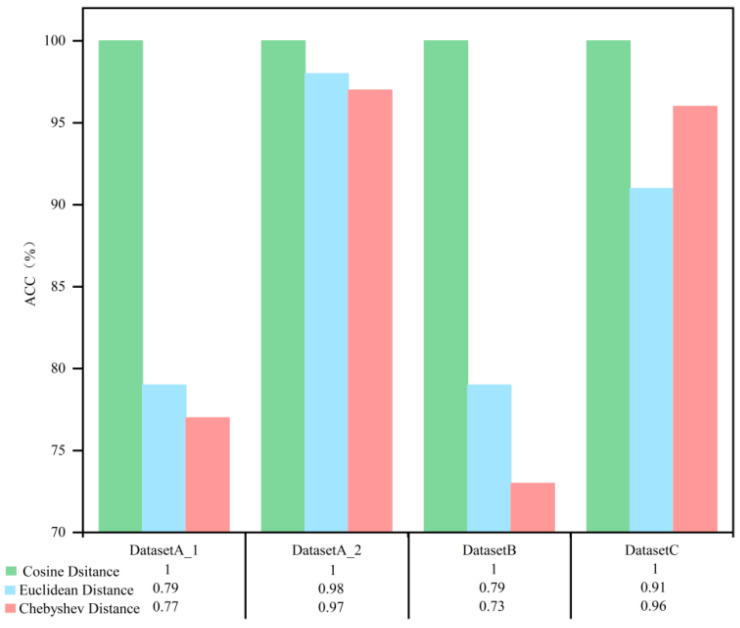
Effect of different distance calculation methods on ACC.

**Table 1 sensors-23-08489-t001:** Test bearing conditions.

Operating Condition	Bearing Dataset	Fault Types	Fault Description	Label
Condition 1	Bearing 1_3	N_1	Normal	0
		OF_1	Outer race fault	1
	Bearing 1_4	CF	Cage fault	2
	Bearing 1_5	OIF	Inner and outer race compound fault	3
Condition 3	Bearing 3_1	N_2	Normal	4
		OF_2	Outer race fault	5
	Bearing 3_2	ICRF	Inner race, cage and rolling body compound failures	6
	Bearing 3_3	IF	Inner race fault	7

**Table 2 sensors-23-08489-t002:** Description of the dataset.

Fault Types	Label	Training Samples			Testing Samples
		Dataset A	Dataset B	Dataset C	Dataset A/B
N_1	0	80%	80%	50%	20%
OF_1	1	80%	80%	30%	20%
CF	2	80%	80%	30%	20%
OIF	3	80%	80%	10%	20%
N_2	4	80%	80%	-	20%
OF_2	5	80%	80%	20%	20%
ICRF	6	80%	80%	10%	20%
ORF	7	80%	80%	20%	20%

**Table 3 sensors-23-08489-t003:** Structure of the model.

Layer	Filter	Nodes	Edges
Input	512	640	5760
GConv_1	512 × 1024	640	5760
BatchNorm_1	1024	640	5760
ReLU_1	-	-	-
GConv_2	1024 × 1024	640	5760
BatchNorm_2	1024	640	5760
ReLU_2	-	-	-
FC_1	1024 × 512	640	5760
Dropout ratio	0.2	640	5760
FC_2	512 × C	640	5760

**Table 4 sensors-23-08489-t004:** DGCN performance on different datasets.

Evaluation Indicators		ACC (%)	TPR (%)	FAR (%)
Dataset				
Dataset A_1		1	98	0.7
Dataset A_2		1	98	0.5
Dataset B		1	98	0.2
Dataset C		1	98	0.2

## Data Availability

The data used for this study is a publicly available dataset linked to: http://biaowang.tech/xjtu-sy-bearing-datasets (accessed on 5 October 2023).

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
