# Peer review of "Intelligent Compound Fault Diagnosis of Roller Bearings Based on Deep Graph Convolutional Network"

_sensors, 2023, doi:10.3390/s23208489_

Round 1
Reviewer 1 Report
The paper conducted research on the identification of bearing faults using intelligent algorithms, which is very meaningful. But there are still some shortcomings in the article:
1. Please clarify which formulas used in the text are existing and which are self-created formulas.
2. There are many different words used for the same words in the text, “cosine similarity” ? or “cosine resemblance”?, and there are Syntax error. Please check the English translation.
3. Add more content to Section 3.2, the current description makes it difficult for readers to understand how graph data is constructed.
4. Add a detailed convolutional network diagram and more descriptions in Section 3.3.
5. There is only condition1 and condition3 in table 1,Where is condition 2?
6. In Section 4.4, 0,1,2,3(Fig.8, 10, 12)…means? Please provide a description.
7. The paper requires more references
The duality of english language needs to be imprved
Author Response
Reviewer #1:
- Please clarify which formulas used in the text are existing and which are self-created formulas.
Response: Equations 17 and 18 are self-created, the rest are extant.
- There are many different words used for the same words in the text, “cosine similarity” ? or “cosine resemblance”?, and there are Syntax error. Please check the English translation.
Response: I have normalized this inconsistent presentation throughout the text. For example “cosine similarity”, “visualization” and so on.
- Add more content to Section 3.2, the current description makes it difficult for readers to understand how graph data is constructed.
Response: I have reorganized the presentation of Section 3.2.
- Add a detailed convolutional network diagram and more descriptions in Section 3.3.
Response: The detailed structure and descriptions has been shown in Figure 2 and 2.3.
- There is only condition1 and condition3 in table 1,Where is condition 2?
Response: There are three working conditions in the original dataset. However, this study mainly focuses on compound faults, which are only included in working condition 1 and working condition 3. Therefore, only these two working conditions are selected in this paper.
- In Section 4.4, 0,1,2,3(Fig.8, 10, 12)…means? Please provide a description.
- The paper requires more references.
Response: Added 2 references (17 and 18) .
Comments on the Quality of English Language: The duality of english language needs to be improved
Response: I have carefully checked for spelling errors and made this correction.
Reviewer 2 Report
This paper proposes a rolling bearing composite fault diagnosis method based on deep graph convolutional network. First, the acquired raw vibration signals is pre-pro-cessed and divided into sub-samples. Secondly, a number of sub-samples in different health states are constructed as graph-structured data, divided into a training set and a test set. Finally, the train-ing set was used as input to a Deep Graph Convolutional Neural Network (DGCN) model, which was trained to determine the optimal structure and parameters of the network.
The research work reported is interesting in the community. Some suggestions are listed below to improve the manuscript's quality (major revision):
1. The manuscript's motivations should be further highlighted in the manuscript, e.g., what problems did the previous works exist? How to solve these problems?
2. The research gaps in the abstract and introduction should be clearly expressed. Please rewrite this part.
3. The authors must clearly explain the difference(s) between the proposed method and similar works in the introduction.
4.The Introduction and/or related work section could be extended and incorporates additional discussions on the topics of advanced techniques, e.g., https://doi.org/10.1016/j.ins.2023.03.142; https://doi.org/10.3390/rs15133402; http://dx.doi.org/10.1145/3513263 and so on.
5. In Figure 2, Data normalization in 1. Data pre-processing includes what?
6. In the section of Case Study, how to determine parameter values?Please provide explanation.
7. What are the limitations behind this study? This topic should be highlighted in the Conclusion of manuscript.
8.The authors are requested to correct all spelling mistakes.
The authors are requested to correct all spelling mistakes.
Author Response
Reviewer #2:
- The manuscript's motivations should be further highlighted in the manuscript, e.g., what problems did the previous works exist? How to solve these problems?
Response: I have highlighted what the problems with the work are and how they can be addressed in the abstract and introduction, which have been highlighted in red throughout the paper.
- The research gaps in the abstract and introduction should be clearly expressed. Please rewrite this part.
Response: These two parts have been rewritten as required.
- The authors must clearly explain the difference(s) between the proposed method and similar works in the introduction.
Response: The main research focus of this paper is how to construct the original vibration signal into graph-structured data. Secondly, the GCN model feature mining capability is utilized to eliminate the screening feature link in fault diagnosis and improve the efficiency of composite fault diagnosis.
- The Introduction and/or related work section could be extended and incorporates additional discussions on the topics of advanced techniques, e.g., https://doi.org/10.1016/j.ins.2023.03.142; https://doi.org/10.3390/rs15133402; http://dx.doi.org/10.1145/3513263 and so on.
Response: Drawing on these three papers, the introduction has been rewritten.
- In Figure 2, Data normalization in 1. Data pre-processing includes what?
Response: Data preprocessing consists of normalizing the raw signal and dividing the subsamples.
- In the section of Case Study, how to determine parameter values?Please provide explanation.
Response: This paper employs the use of momentum stochastic gradient descent (SGD) as an optimiser for hyperparameter optimization.
- What are the limitations behind this study? This topic should be highlighted in the Conclusion of manuscript.
Response: However, the study was not integrated with the bearing mechanism, and therefore, the interpretability is not strong. In addition to this, this study uses a standard data set, which has little ability to generalize to actual operational data.
- The authors are requested to correct all spelling mistakes.
Response: I have carefully checked for spelling errors and made this correction.
Reviewer 3 Report
The paper is nicely written and the topic is well explained. It is an interesting approach to make use of GCN in fault diagnosis. Basically, the idea is not new, but the authors manage to show its applicability and make it easier for interested readers to use it for similar problems.
Just a question regarding the method applied. As you used well defined fault cases for training and test, your network is able to predict the type of fault. Have tested to introduce a completely new fault type which was not seen by the network before?
Thank you!
The authors need to check thoroughly for language and grammar issues. Right in the title it must read based on instead of base on. Furthermore, the sentence structure of the following need to be revised as four bearing types in the end is not correct.
Dataset A contains two conditions, Dataset A_1 includes normal, outer ring fail-ure, cage failure, combined inner and outer ring failure four bearing types. Dataset A_2 includes normal, outer ring failure, combined inner ring, rolling element and cage failure, inner ring failure four bearing types.
Author Response
Reviewer #3:
- Just a question regarding the method applied. As you used well defined fault cases for training and test, your network is able to predict the type of fault. Have tested to introduce a completely new fault type which was not seen by the network before?
Response: Compound faults are relatively rare, so this study did not introduce an entirely new fault type for testing.
- The authors need to check thoroughly for language and grammar issues. Right in the title it must read based on instead of base on. Furthermore, the sentence structure of the following need to be revised as four bearing types in the end is not correct.
Dataset A contains two conditions, Dataset A_1 includes normal, outer ring failure, cage failure, combined inner and outer ring failure four bearing types. Dataset A_2 includes normal, outer ring failure, combined inner ring, rolling element and cage failure, inner ring failure four bearing types.
Response: I have carefully checked for spelling errors and made this correction.
The above statement is amended to read: Dataset A contains two working conditions, part of the fault types are normal, outer ring failure, cage failure and inner and outer ring composite failure. The other part of the fault types are normal, outer ring failure, inner ring, rolling element and cage composite failure, and inner ring failure.
Reviewer 4 Report
It seems to be an interesting application of weighted graphs but it is more a feature extraction development rather than deep learning – the combined use of both approaches is not new but results are interesting.
-In the title the word “base” shouldn’t be “based”?
- The first phrase in the abstract needs to be reformulated since it seems to be incomplete!
- In the abstract “the training set was used”, I suggest the use of “is used”
- In the abstract it is stated “….provides a new idea”, perhaps “…. Provides a new approach”
- In page 1 the phrase “To overcome the above shortcomings, deep learning theory is introduced into the field of fault diagnosis.” Should say a bit more! .. Or perhaps eliminated since it does not add much!
- In page 2 it is stated that “Compared with the traditional shallow model, deep learning builds an "end-to-end" model that does not rely on prior knowledge.” – it does in fact rely on prior knowledge! It is not in the traditional way, but it is there! Please reformulate!
- In page 2 it is stated that “But the above deep learning method can only learn features from the vertices of the input data, neglecting the information contained in the edges formed between the vertices.” – What do you mean by vertices?! A small sample of many other deep learning approaches should be acknowledged in this work.
- In page 2 it is stated that “This method improves the fault diagnosis efficiency and provides a new idea for bearing com-pound fault diagnosis route.” – please provide background work evidences – improves what in specific?.
- End of Page 4 it is stated “use the resulting spectrum as a new sample,…” – please rephrase since it might be confusing, there is in fact a set of features after FFT, or otherwise transformed sample. Does this new sample preserve amplitude and phase?
In page 5 it is stated that “BatchNorm layer makes edge and node training faster and more stable” – provide some reference to support that claim.
In the description of the experimental setup there is no sampling frequency? Check it. You need to specify the RPM under which were the experiments carried out. It so, also which loads were tested.
Fig. 3 under point 4.2 is not correctly placed.
In pag. 8 it is stated “Each layer of the network is visualized by Gephi,..” – what is Gephi? … might seem common but may not be known. Is it really necessary to reference Gephi?
Minor changes/revision required.
Author Response
Reviewer #4:
-In the title the word “base” shouldn’t be “based”?
- The first phrase in the abstract needs to be reformulated since it seems to be incomplete!
Response: Modified to: The high correlation between rolling bearing composite faults and single fault samples is prone to misclassification.
- In the abstract “the training set was used”, I suggest the use of “is used”
- In the abstract it is stated “….provides a new idea”, perhaps “…. Provides a new approach”
Response: Modified to:The experimental results show that the DGCN can effectively identify compound faults in rolling bearings, which provides a new approach for the identification of compound faults in bearings.
- In page 1 the phrase “To overcome the above shortcomings, deep learning theory is introduced into the field of fault diagnosis.” Should say a bit more! .. Or perhaps eliminated since it does not add much!
- In page 2 it is stated that “Compared with the traditional shallow model, deep learning builds an "end-to-end" model that does not rely on prior knowledge.” – it does in fact rely on prior knowledge! It is not in the traditional way, but it is there! Please reformulate!
Response: Modified to: Compared with the traditional shallow model, deep learning builds an "end-to-end" model.
- In page 2 it is stated that “But the above deep learning method can only learn features from the vertices of the input data, neglecting the information contained in the edges formed between the vertices.” – What do you mean by vertices?! A small sample of many other deep learning approaches should be acknowledged in this work.
Response: Modified to: However, the deep learning methods described above can only learn their own node features, ignoring the information contained in the edges formed between neighboring nodes.
- In page 2 it is stated that “This method improves the fault diagnosis efficiency and provides a new idea for bearing com-pound fault diagnosis route.” – please provide background work evidences – improves what in specific?.
Response: Modified to: This study provides a new approach to the composite fault diagnosis route for bearings.
- End of Page 4 it is stated “use the resulting spectrum as a new sample,…” – please rephrase since it might be confusing, there is in fact a set of features after FFT, or otherwise transformed sample. Does this new sample preserve amplitude and phase?
Response: Modified to: Therefore, we perform a Fast Fourier Transform (FFT) on each subsample and use the transformed data as a new sample, the process can be expressed as:
In page 5 it is stated that “BatchNorm layer makes edge and node training faster and more stable” – provide some reference to support that claim.
-In the description of the experimental setup there is no sampling frequency? Check it. You need to specify the RPM under which were the experiments carried out. It so, also which loads were tested.
Response: Added: The sampling frequency was 25.6 kHz and the sampling -interval was 1 min. The experiments selected in this paper are shown in Table 1. In this paper, six bearings were selected for two operating conditions, with a speed of 2100 r/min for condition 1 and 2400 r/min for condition 3.
Fig. 3 under point 4.2 is not correctly placed.
-In pag. 8 it is stated “Each layer of the network is visualized by Gephi,..” – what is Gephi? … might seem common but may not be known. Is it really necessary to reference Gephi?
Comments on the Quality of English Language: Minor changes/revision required.
Response: I have carefully checked for spelling errors and made this correction.
Round 2
Reviewer 2 Report
Moderate editing of English language required
Moderate editing of English language required